# Presence of Perfluoroalkyl Substances in Landfill Adjacent Surface Waters in North Carolina

**DOI:** 10.3390/ijerph20156524

**Published:** 2023-08-04

**Authors:** Aleah Walsh, Courtney G. Woods

**Affiliations:** Department of Environmental Sciences and Engineering, Gillings School of Public Health, University of North Carolina at Chapel Hill, Chapel Hill, NC 27599, USA; arwalsh@live.unc.edu

**Keywords:** PFAS, leachate, emerging contaminants, industrial sludge

## Abstract

Landfills pose an important public health risk, especially in historically disenfranchised communities that are disproportionately sited for landfills and in rural areas where private wells may be impacted. Landfills are major sources of perfluoroalkyl substances (PFAS) that migrate into the surrounding environment. This study characterized PFAS in surface waters adjacent to two landfills, one in Sampson County (SC) and one in Orange County (OC) in North Carolina. In addition to municipal solid waste and construction and demolition waste, the landfill in SC accepts industrial sludge from a chemical plant that produces proprietary PFAS. Over four months, 35 surface water samples were collected at upstream, landfill-adjacent, and downstream/downgradient sites. Thirty-four PFAS were analyzed using liquid chromatography with tandem mass spectroscopy. Of those, six novel and six legacy PFAS were detected. Legacy PFAS were detected in surface water near both landfills, with the highest concentrations adjacent to the landfill. Novel PFAS were only detected in surface water near the SC landfill and showed the highest concentrations adjacent to the landfill, indicating offsite migration of PFAS. These findings support the need for more comprehensive and frequent monitoring of groundwater and surface water wells near landfills and stricter regulation regarding the landfilling of industrial materials.

## 1. Introduction

Perfluoroalkyl substances (PFAS) are a class comprised of almost 5000 chemical compounds, many of which are used in a wide range of consumer products. These synthetic chemicals have gained attention over the last decade due to their persistence in the body and in the environment [1]. Although toxicity data for PFAS in humans is still emerging, PFAS are associated with negative health outcomes, including increased risk of cancer, adverse birth outcomes, and changes in liver enzymes [2,3,4,5,6].

PFAS gained further attention in North Carolina when high levels of these compounds were measured in the Cape Fear River, which runs adjacent to Chemours, a facility manufacturing PFAS-containing products [7,8,9,10]. In addition to detecting elevated levels of previously characterized PFAS (i.e., legacy) compounds, researchers also discovered a novel chemical, hexafluor- opropylene oxide dimer acid (HFPO-DA), or “GenX”, in the Cape Fear River. Following this discovery, a long list of novel compounds were identified in the Cape Fear River and at outfalls from the Chemours facility, including GenX, Nafion byproduct 2 (Nafion BP2), Nafion byproduct 1 (Nafion BP1), Nafion byproduct 4 (Nafion BP4), Perfluoro(3,5,7-trioxaoctanoic) acid (PFO3OA), Perfluoro(3,5,7,9-tetraoxadecanoic) acid (PFO4DA), Perfluoro(3,5-dioxahexanoic) acid (PFO2HxA), Perfluoro-3,5,7,9,11-pentaoxadodecanoic acid (PFO5DoA), Perfluoroethoxypropyl carboxylic acid (PEPA), Perfluoromethoxypropyl carboxylic acid (PMPA), Perfluoroethoxysulfonic acid (NVHOS), and Perfluoroethoxsypropanoic acid (Hydro-EVE). As of 2022, the Chemours facility will dispose of their industrial sludge at the Sampson County landfill, as listed on their National Pollution Discharge Elimination System (NPDES). In 2015, they sent approximately 35,000 pounds of solid waste to the Sampson County landfill every week [11].

Previous studies have found legacy PFAS at elevated levels in leachate from municipal solid waste (MSW) landfills [12,13,14,15,16,17,18]. The efficacy of traditional landfill liners for preventing PFAS from entering the environment is not well known, and PFAS concentrations in surface water or groundwater wells are not currently monitored by landfills in NC. 

This study aims to characterize PFAS in landfill-adjacent surface waters at two locations in North Carolina: the Sampson County landfill in Roseboro, NC, and the Orange County landfill in Chapel Hill, NC. Orange County was selected as another site with no known history of industrial sludge disposal that could provide an understanding of the baseline municipal solid waste (MSW) PFAS profile. 

This work provides insight into the potential exposure risk posed by the presence of PFAS measured at these two sites. The presence of potentially toxic compounds such as PFAS in elevated concentrations in surface water could indicate a threat to nearby private wells, thereby warranting monitoring of surface and groundwater for these compounds at the landfill and in private wells.

## 2. Materials and Methods

**Landfill selection.** The Sampson County landfill, operated by Waste Industries Inc., is located off Hwy-24 near the rural community of Snow Hill, between Roseboro and Clinton. The 1315-acre site includes a closed municipal solid waste (MSW) landfill, a closed construction and demolition (C&D) landfill, an active MSW landfill, and an active C&D landfill. This landfill is the largest in North Carolina and receives waste from several transfer stations across the state [19]. Both the closed and active MSW landfills are lined, although it is unclear whether they are lined with a geomembrane or clay lining [20,21]. The C&D landfills are not lined. It is not publicly reported which section of the landfill received the industrial sludge from Chemours.

Bearskin Swamp Creek runs adjacent to these landfills, flowing from North to South, and is approximately 560 ft from the landfill edge. This is a wetland area of relatively low elevation located in the coastal plains of North Carolina. There is also a small African-American community, Snow Hill, of approximately 20 households as close as 200 ft northwest of the inactive landfill edge.

The Orange County landfill, a 220-acre site, is located in northern Chapel Hill off of Hwy 86, between Rogers Road and Eubanks Road [22]. This landfill is comprised of two closed MSW landfills (one lined, one unlined), one active C&D landfill (unlined), one inactive C&D landfill (unlined), and a household hazardous waste (HHW) landfill (above ground) [19]. Old Field Creek runs adjacent to the landfill between the southern MSW and C&D sites and is approximately 200 feet from the landfill edge. There is also a historically African-American community, the Rogers-Eubanks community, of approximately 50 households as close as 1500 ft south of the landfill edge.

**Sampling site selection.** In Sampson County, sampling sites were identified along the continuous stream of Bearskin Swamp Creek. These sites are shown on the map in Figure 1. The upstream site was located near Halls Pond (HP). This site was sampled directly after Halls Pond, where Bearskin Swamp Creek intersects with Bonnetsville Road, approximately 3 miles upstream of the landfill. Sampling sites proximal to the landfill were along Mitchell Loop Road (ML), which is approximately 1500 ft from the outermost edge of the current C&D portion of the landfill and 2000 ft from the active MSW landfill. Site ML is along Bearskin Swamp Creek, just upstream of the tributary that intersects at Mitchell Loop Rd. This site is approximately 1500 ft from the active C&D landfill. The Roseboro Highway (RB) sampling site is located where Roseboro Highway intersects with Bearskin Swamp Creek, approximately 1000 ft from the outermost edge of the closed municipal solid waste landfill (permit number 82–01) [20]. The McLemore Road site (MK) is located approximately 1500 ft downstream from the closed municipal solid waste landfill, just above where the tributary crossing McLemore Rd joins Bearskin Swamp Creek. The McLemore Tributary (MKt) site is located in the tributary that crosses McLemore Rd and was sampled before the tributary joins Bearskin Swamp Creek. The Keith Road (KR) site is a small tributary to Bearskin Swamp Creek fed by a pond downgradient from the landfill but is not a part of the continuous stream. The Fleet Cooper (FC) site is located where Little Coharie Creek intersects Fleet Cooper Road downstream of the landfill after Bearskin Swamp Creek has joined Little Coharie Creek. 

In Orange County, continuous stream sampling sites were identified along Old Field Creek. Additional sites downgradient of the landfill but not along the continuous stream were also identified. Accessing a site upstream of the landfill along Old Field Creek was difficult due to private property restrictions. Additionally, during usual hydraulic conditions, the headwaters are proximal to the landfill. Therefore, Old Field Creek was sampled in only two locations: one proximal to the landfill and one downstream. A map of the sampling sites is shown in Figure 2. The Millhouse Road (MH) site is located where Old Field Creek intersects Millhouse Road, approximately 600 ft from the closed MSW landfill. The Clyde Road (CR) site is located where Old Field Creek intersects Highway 86 near Clyde Road, approximately one mile downstream from the MH site. Though the site is north of the landfill, it is downstream of the landfill. In addition to these sites along Old Field Creek, two additional sites were sampled. The Tallyho (TH) site is located where a tributary of Bolin Creek intersects Tallyho Trail. The Winmore (WM) site is located farther downstream, where Bolin Creek intersects with East Winmore Avenue. The TH and WM sites are downgradient from the landfill but are not part of the continuous stream. These sites were sampled to detect the possible influence of the landfill on surface water via groundwater influence, runoff, or percolation within a greater distance from the landfill. The latitude and longitude of each site are supplied in Table 1. 

**Sampling materials.** One-liter Nalgene^TM^ high-density polyethylene (HDPE) bottles from ThermoFischer Scientific were used for collecting samples in the field. All HDPE bottles were acid washed with trace metal-grade hydrochloric acid (HCL) from Fischer Scientific instead. Bottles were filled with 3M HCL and left to sit for approximately 24 h, rinsed with deionized (DI) water five times, then filled with DI water and soaked for another 24 h. Bottles were given a final series of washes (3x) with DI water before being brought to the field. The bottles were reused but underwent acid washing between samples. MilliporeSigma^TM^ Swinnex^TM^ Filter holders are comprised of polypropylene casings and a small silicone o-ring. The filter holder casings underwent the same acid washing process as described above for the HDPE bottles. The silicon o-rings were soaked in DI water for 24 h, rinsed five times, soaked for another 24 h, and rinsed three more times between uses. Samples were filtered into 500 mL Thermo Scientific^TM^ Nalgene^TM^ HDPE bottles, which also underwent the same acid washing process as the 1L bottles. However, the 500 mL HDPE bottles were not reused. Exel International polypropylene syringes from Thermo Scientific^TM^ were single-use and not acid-washed before use.

**Field methods.** Samples were collected at each sampling location approximately once per month from October 2019 to January 2020. All field samples were collected in acid-washed 1L HDPE bottles. Bottles were rinsed three times with sample water, then filled.

**Sample analysis.** Samples were filtered upon return to the laboratory and within 8 h of collection. Five hundred mL of sample was filtered using Whatman^TM^ GF/A glass microfiber filters and stored at room temperature in clean, acid-washed HDPE bottles. When filtering, the syringe barrel was rinsed with the sample three times before passing through the filter. The first 10 mL were passed through to waste in order to rinse the filter with the sample prior to collection. The 500 mL HDPE bottles were rinsed with approximately 20 mL of filtered water three times before collection.

**Analytical methods.** As described previously, due to the known disposal of PFAS-containing sludge at the Sampson County Landfill and the likely presence of PFAS through the disposal of consumer products, we chose to measure a number of compounds that are proprietary to Chemours, henceforth referred to as novel PFAS, including GenX, Nafion BP2, Nafion BP1, Nafion BP4, PFO3OA, PFO4DA, PFO2HxA, PFO5DoDA, PEPA, PMPA, NVHOS, and Hydro-EVE, as well as several other PFAS commonly present in consumer products and, thus, landfill leachates, including PFHpA, PFOA, PFNA, PFDA, PFHxS, PFOS, PFBA, PFPeA, PFBS, and PFHxA. Additionally, Perfluorododecanoic acid (PFDoA), Perfluorododecanesulfonic acid (PFDoS), Perfluorodecanesulfonic acid (PFDS), Perfluoroheptanesulfonic acid (PFHpS), Perfluorohexadecanoic acid (PFHxDA), Perfluorononanesulfonic acid (PFNS), Perfluorooctadecanoic acid (PFODA), Perfluorooctanesulfonamide (PFOSA), Perfluoropentanesulfonic acid (PFPeS), Perfluorotetradecanoic acid (PFTA), Perfluorotridecanoic acid (PFTrA), and Perfluoroundecanoate acid (PFUnA). 

PFAS concentrations were analyzed using liquid chromatography with tandem mass spectrometry (LC-MS) in accordance with the large volume direct injection method described by Roberts et al. with SCIEX laboratories [23]. Samples were analyzed on a Sciex 6500 triple-quadrupole MS using an Agilent ZORBAX Eclipse Pls C18 3.0 × 50 mm analytical column. To prepare for injection, samples were poured from the 500 mL storage bottle into a graduated cylinder so that sample bottles could be rinsed with methanol to mobilize any PFAS that may have adsorbed to the sample bottle. Each sample was then poured back into its respective 500 mL HDPE bottle and shaken thoroughly. One mL of sample was then filtered into a Microsolv reduced surface activity (RSA) total recovery clear glass vial using a 10 mL syringe and a Whatman glass fiber 0.70 μm 25 mm filter in polypropylene filter holders. Prior to filtering, the 10 mL syringe barrels were rinsed with methanol. Filters were rinsed with 3 mL of methanol followed by 3 mL of sample water before use. Samples were then spiked with 50 μL of internal standard and 50 μL of methanol before loading on the mass spectrometer. Filter holders were cleaned with methanol prior to use.

Procedural blanks were prepared using DI water. These blanks underwent the exact same methods as described for samples to account for any potential contamination from the method materials. Two duplicate samples were included to ensure consistency in the values obtained from the mass spectrometer.

T-tests were also conducted to determine whether differences observed between the reference site and down gradient sites were statistically significant and whether the difference between the procedural blank and the reference site was significant.

**Limit of detection and limit of quantification.** PFAS results were processed using MultiQuant software from SCIEX. The limit of detection (LOD) varies by compound due to differences in ionization efficiency. To qualify as above-LOD for the purposes of this study, the peaks had to meet the following criteria: (1) the signal-to-noise ratio (S/N) had to account for ≤1% of total error as calculated by Equation (1), (2) the intensity of the peak (measured by height) had to be, at minimum, of the same order of magnitude as the lowest standard (0.02 ppb), (3) the calibration curve correlation coefficient had to be at least 0.99, and (4) upon visual inspection, the peak shape needed to closely resemble that of the standards. It should be noted that the limit of quantification (LOQ) for the purposes of this study is the value of the lowest standard, 20 parts per trillion (ppt).
(1)$${\mathrm{Error}}_{S/N}\;=\frac{50}{S/N}$$

Equation (1): This equation estimates the percent error attributable to uncertainty from the signal-to-noise ratio [24].

## 3. Results

**Cumulative PFAS.** Thirty-five samples collected between October 2019 and January 2020 from sampling sites near Sampson and Orange County landfills were analyzed for 34 PFAS compounds. Of the 34 PFAS compounds analyzed, twelve were determined to have peaks above the detection limit. The twelve compounds with qualifying peaks include: PFBA, PFPeA, PFBS, PFHxA, PFHpA, PFOA, GenX, PMPA, PEPA, NVHOS, Nafion BP2, and Nafion BP4. As shown in Figure 3a,b, the sum of the concentrations of these twelve PFAS ranged from 33.3 ppt to 780 ppt. 

The most prominent PFAS species measured in Sampson County were Nafion BP4, PFPeA, and PFBA. PFAS concentrations were lowest at the upstream site (HP) and the tributary sites (MKt, KRt) and highest at the sites proximal to the landfill (ML, RB, and MK). Concentrations well downstream of the landfill at FC are generally higher than HP, but not as high as those measured proximal to the landfill.

Figure 3 also shows PFAS measurements observed in Orange County. The most prominent PFAS species were PFHxA, PFPeA, PFBA, and PFOA. PFAS concentrations were highest at MH, the site proximal to the landfill. Concentrations were slightly lower downstream at CR. At sites not along the continuous stream, concentrations were generally lower compared to MH. The lowest concentrations were measured at WH, the site furthest from the landfill.

**Novel PFAS.** The novel PFAS (GenX, PMPA, PEPA, NVHOS, Nafion BP2, and Nafion BP4) are shown individually in Figure 4. 

All novel PFAS concentrations measured in this study are negligible upstream of the landfill in Sampson County and in the sampled tributaries. The concentrations measured at the landfill-proximal sites are all elevated. The concentrations in Orange County are at least one order of magnitude lower than the concentrations measured in Sampson County. Furthermore, concentrations measured in Orange County do not exhibit any difference with respect to landfill proximity. All novel PFAS in Sampson County, with the exception of Nafion BP2 and PEPA, are above the LOQ at landfill-proximal sites. In Orange County, however, none of the novel PFAS are above the LOQ.

**Legacy PFAS.** Concentrations of the legacy PFAS (those that have been identified previously in landfill leachate) are shown in Figure 5. PFBS, PFHxA, PFHpA, and PFOA are all detected at elevated levels proximal to the landfill, with concentrations decreasing downstream of the landfill. This trend was observed in Sampson County and Orange County. In Sampson County and Orange County, all legacy PFAS, with the exception of PFBS and PFHpA, are above the LOQ at landfill proximal sites.

**Fold change.** Table 1 shows the upstream watershed area for each sampling location. In Sampson County, concentrations of PFBA, PFHxA, and NVHOS at landfill proximal sites ML and RB are more than ten times higher than at HP. These differences are statistically significant (with a *p*-value less than 0.05 when performing a t-test) in Sampson County at sites abbreviated as ML and RB for most compounds. The difference is not significant at MKt for any compounds. FC is statistically different from HP for PFPeA, PFOA, and Nafion BP4. Several novel PFAS, including PFPeA, Nafion BP4, and GenX, are all approximately 20 times higher at landfill-adjacent sites compared to upstream sites. In Orange County, the concentrations of PFBA are statistically different from those of MH at all other sites. For all other compounds, CR and TH are not significantly different from MH. The concentrations of all compounds, excluding PFBS, PFHxA, NVHOS, and GenX, are significantly different at WM compared to MH. Concentrations of PFBA, PFHxA, and PFHpA are all more than ten times higher at MH compared to WM. 

## 4. Discussion

The concentrations of the upgradient to downgradient profiles of the legacy PFAS were similar at the Orange and Sampson County landfills, with the exception of PFPeA and PFBA. This may suggest that the legacy compounds are indicative of landfill leachate enriched in PFAS from disposed cookware, treated fabrics, and other regular household waste. While PFPeA and PFBA have been identified in landfill leachate, they appear to be more elevated at the Sampson County sites. Although these compounds are not novel, it is possible that they may be associated with the industrial production of PFAS in North Carolina [26].

Our results indicate that PFAS concentrations at both locations were generally highest at the sites proximal to the landfill, lower downstream, and lowest upstream (in cases where upstream sites could be accessed). This trend is most notable at Sampson County sites, where PFAS levels range from 0.85- to 26.2-fold higher at landfill-proximal sites compared to upstream sites. The novel PFAS compounds NVHOS, Nafion BP4, and GenX exhibited statistically significant differences and were 10.21, 19.85, and 19.24-fold higher, respectively, near the landfill as compared to upstream. Concentrations of novel PFAS were also detected at lower concentrations at sites farther downstream and in the tributaries downstream. These findings lend strong support for our hypothesis that contamination (via landfill leachate or runoff) is entering the surrounding environment.

One previous study utilized PFAS as an effective tracer of landfill leachate in groundwater, hypothesizing that the PFAS found in the effluent were residual from fabrics, insulation, firefighting foam, carpets, cookware, and other household items discarded in the landfill [27]. Other studies have measured PFAS in both groundwater and landfill leachate, after which they were able to infer the influence of the landfill on natural waters [14,15,16,17,18]. This study, to the best of the authors’ knowledge, is the first to measure surface water levels of PFAS as a way to specifically determine off-site migration of landfill leachate. This study is also the first to analyze the extent of novel PFAS contamination in surface waters proximal to a landfill accepting industrial sludge from a facility that produces PFAS-containing chemicals.

**Limitations.** This study was limited by a small budget and by the amount of time over which sampling was conducted. Since contiguous samples were not collected at all sites at the Sampson County location on every sampling day, this study relies heavily on the assumption that temporal/seasonal variability are low. Another significant limitation is the small number of samples collected at sampling site HP, located upstream of the Sampson County landfill. Given this preliminary data, however, it is apparent that the stream area between HP and ML is critical to understanding potential landfill influence. Another limitation is that data were not collected on flow rate, which can be a helpful metric when calculating load and normalizing results across sampling locations. Due to this being a pilot study with limited field sampling equipment, flow rates were not collected. Lastly, though this study did not observe general trends between concentration and distance from the landfill, the ability to assert the source of the contamination is limited since a true upstream site in Orange County was not sampled due to inaccessibility.

A minor limitation of this study is that the hydrology, underlying geology, and landfill characteristics (i.e., age, size, waste composition, lining, etc.) vary between Sampson County and Orange County. Therefore, comparisons between landfills must be made with caution. The watershed area was determined using the Watershed Tool in ArcGIS Pro. Table 1 shows the upstream watershed area at each site. This data provides a proxy for relative stream size. Sites with a larger upstream watershed area will take in water from a larger area; therefore, both the load of a particular contaminant and the total volume of water are likely to be greater. This also, therefore, serves as a proxy for relative flow between sites. However, depending on the relative contribution of runoff from a particular contaminant, a larger watershed area may mean a higher load or a greater dilution effect. Precipitation likely also impacts the concentration of the analytes included in this study, though there are not enough sampling events to identify a trend. 

Venkatesan and Halden (2013) report that PFAS are consistently present in biosolids and sewage sludge from wastewater treatment plants (WWTP) [28]. These biosolids are occasionally stored in landfills and occasionally applied on agricultural lands. In North Carolina, land application of sewage and WWTP sludge is a common practice. Given that Sampson County is a rural county with thousands of acres of agricultural land, including the second highest density of swine farms in the state, land application of WWTP sludge is possibly a source of PFAS contamination into waterways such as Bearskin Swamp Creek. All samples collected at upstream site HP and tributary sites MKt and KRt were not measured above the LOQ, and thus conclusions regarding the significance of PFAS measured at these sites cannot be drawn.

**Community health implications and recommendations to regulatory agencies.** Historically, landfills in North Carolina are disproportionately located in rural, low-income communities of color, many of which rely on private wells for drinking water [29]. These communities may have limited capacity to test their water regularly, and the cost of installing water filtration systems (such as reverse osmosis systems) can be prohibitive.

Though the study assesses raw surface water samples, it is worth noting that the cumulative concentrations measured in this study exceed the action levels that existed at the time of the study (70 ppt for combined select PFAS or 10 ppt for any individual select PFAS) established in a 2019 consent order between the Department of Environmental Quality of North Carolina and Chemours. Since then, the EPA has established health advisory limits of 0.004 ppt for PFOA, 0.02 ppt for PFOS, 10 ppt for GenX chemicals, and 2000 ppt for PFBS. They have also proposed a minimum contaminant level of 4 ppt for PFOA and PFOS [30]. Based on the preliminary findings of this study, there is a possible threat posed to recreational and groundwater resources proximal to the Sampson County landfill. Given the presence of novel PFAS in surface water observed in this study, it is possible they may be present in groundwater as well. If novel PFAS are found in well water in Sampson County, remediation efforts similar to those outlined in the consent order should be instituted to protect public health.

## 5. Conclusions

This study, to the best of the authors’ knowledge, is the first to measure surface water levels of PFAS as a way to specifically determine off-site migration of landfill leachate. This study is also the first to analyze the extent of novel PFAS contamination in surface waters proximal to a landfill accepting industrial sludge from a facility that produces PFAS-containing chemicals.

The presence of PFAS at the landfill adjacent sites at both case study locations suggests that there is potential migration of landfill leachate off site. The elevated levels of novel PFAS at sites proximal to the landfill in Sampson County provide compelling evidence to support the recommendation for monitoring of PFAS chemicals at the landfill during biannual monitoring events and reporting to state and federal officials. Furthermore, households in this area that rely on private wells may need to be tested.

Furthermore, monitoring events in surface and groundwater should occur with more regularity. In addition to increasing the frequency of monitoring events, the allowable distance between residences and landfills should also be reconsidered. The results of this study raise the important question of whether industrial sludge can be disposed of at MSW and/or C&D landfills. Therefore, we suggest that regulators conduct further research into the types of waste accepted at MSW and C&D landfills and the requirements for industrial sludge management. 

## Figures and Tables

**Figure 1 ijerph-20-06524-f001:**
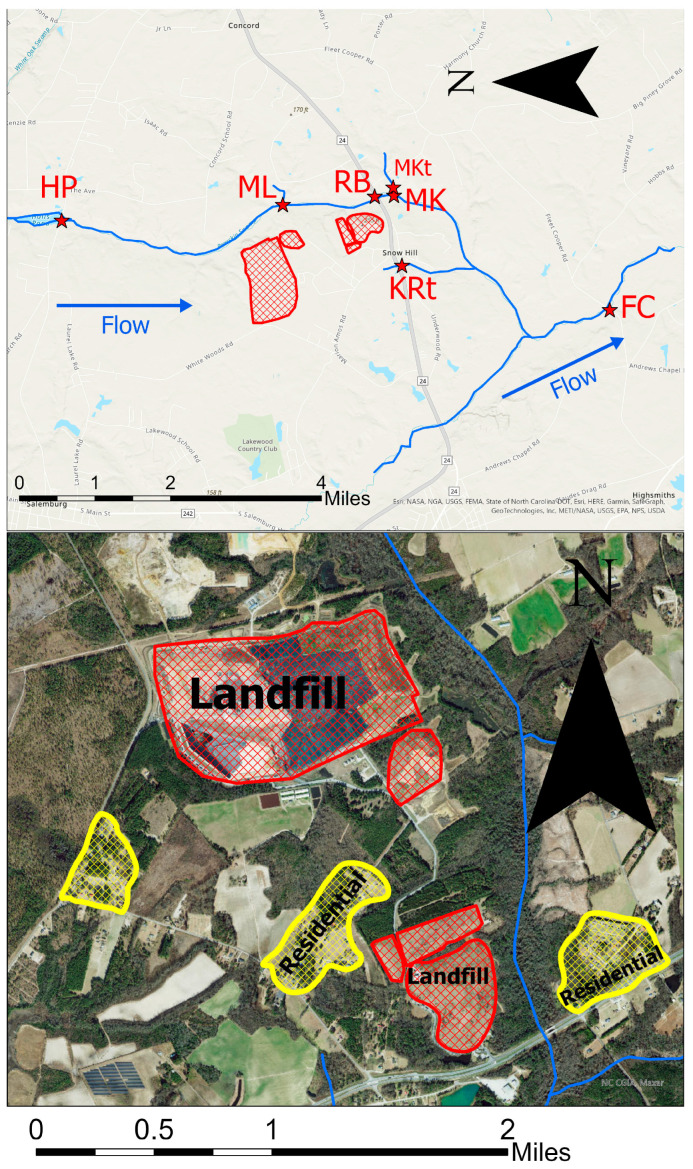
Map of Sampson County landfill with water sampling sites (**top**). Landfills are shown in red. Satellite image of Sampson County landfill (**bottom**). Landfills are shown in red, and residential areas are shown in yellow. Maps were prepared in ArcGIS Pro on 23 July 2023.

**Figure 2 ijerph-20-06524-f002:**
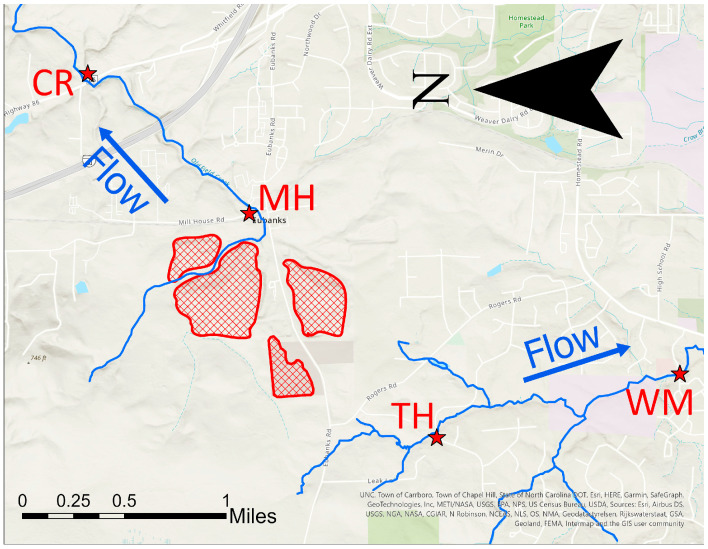
Map of the Orange County landfill with water sampling sites. Landfills are shown in red. This map was prepared in ArcGIS Pro on 30 July 2023.

**Figure 3 ijerph-20-06524-f003:**
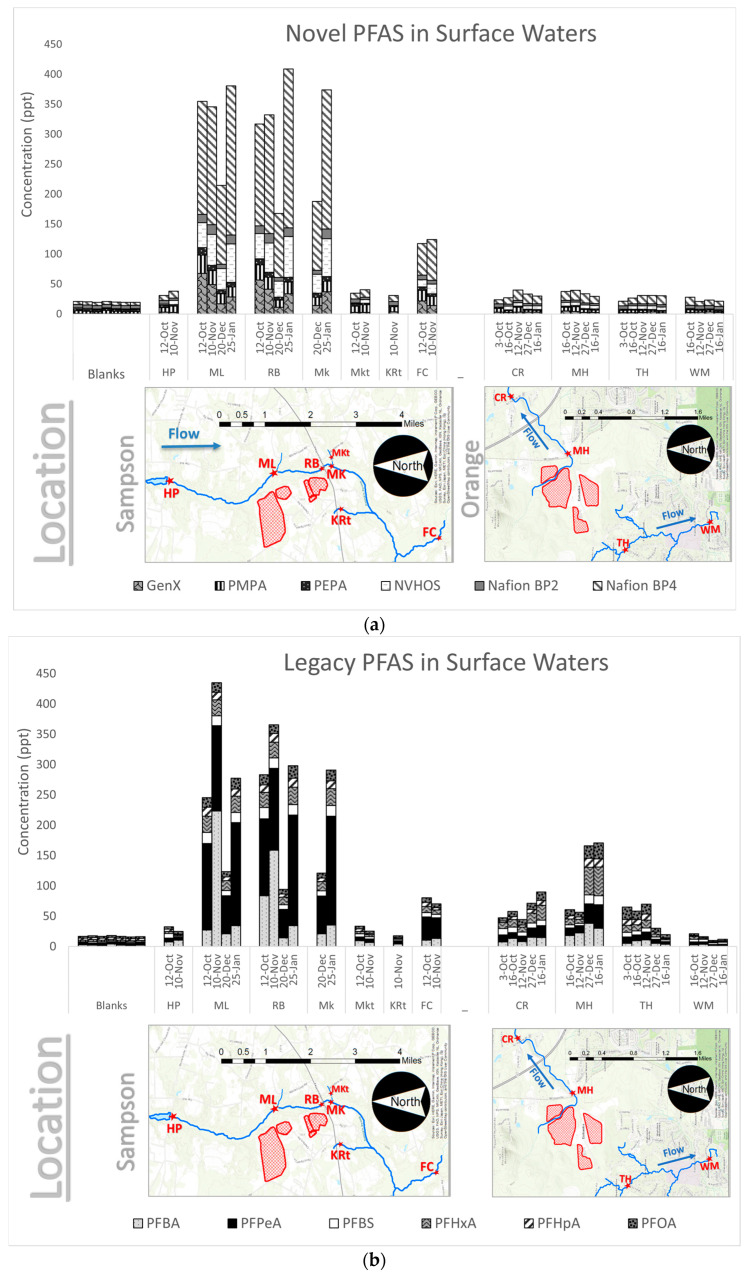
(**a**). Cumulative PFAS concentrations for six Novel PFAS species above the LOD with map overlay. Dates are shown in day–month format. (**b**). Cumulative PFAS concentrations for six Legacy PFAS species above the LOD with map overlay. Dates are shown in day–month format. Each bar represents one sample per site, per day. Note that the scale bars for the Orange County and Sampson County maps shown beneath the plots are different. Procedural blanks are shown to validate that sample integrity was maintained by the method. (Figure adopted from Walsh, 2020 [25]). Landfills are shown in red.

**Figure 4 ijerph-20-06524-f004:**
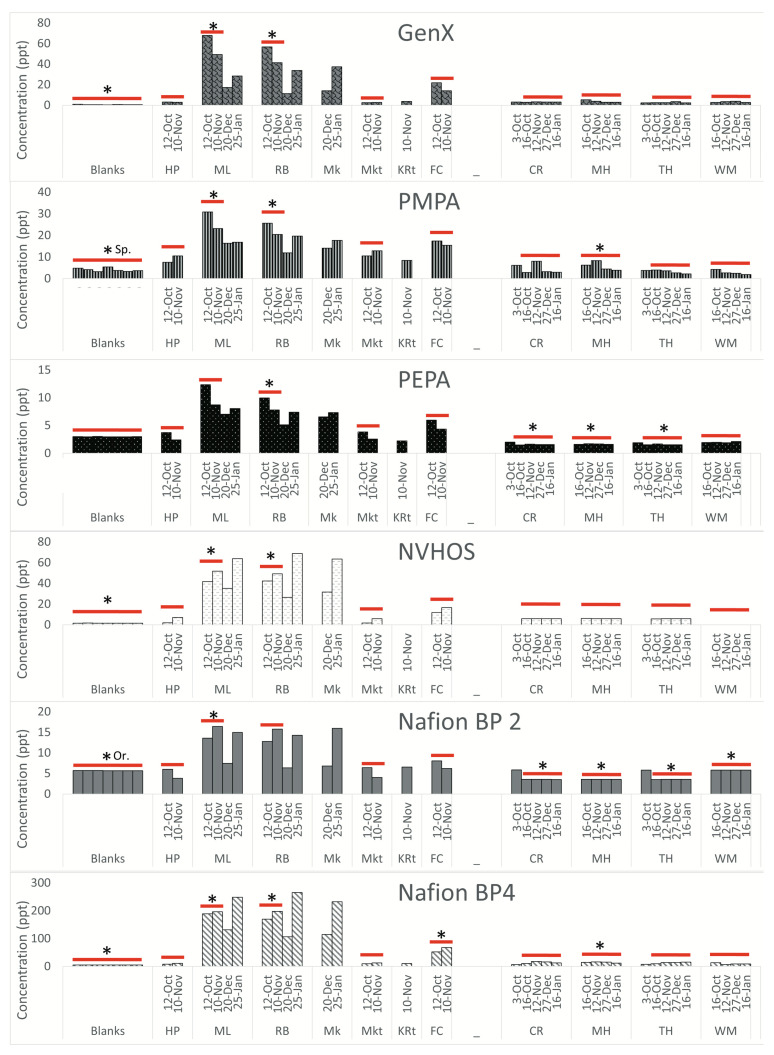
Novel PFAS compounds. Procedural blanks are shown to validate that sample integrity was maintained by the method. The LOQ for all compounds is 20 ppt. In Sampson County, fold change was calculated by averaging the concentration for samples collected on 12 October and 10 November and dividing by the average concentration of the reference site samples from 12 October and 10 November. In Orange County, fold change was averaged using samples collected on 16 October, 12 November, 27 December, and 16 January. Samples from these dates were selected for this calculation because we had samples from almost all of the sampling sites these dates. Samples included in these calculations are indicated by the red line. Statistical significance (a *p*-value less than 0.05 when performing a *t*-test) is indicated by an asterisk. If the blanks were statistically significantly different than the reference site in only one county, the significant county abbreviation is included. The reference sites were HP in Sampson County and WM in Orange County. Dates are shown in day–month format. (Figure adopted from Walsh, 2020 [25]).

**Figure 5 ijerph-20-06524-f005:**
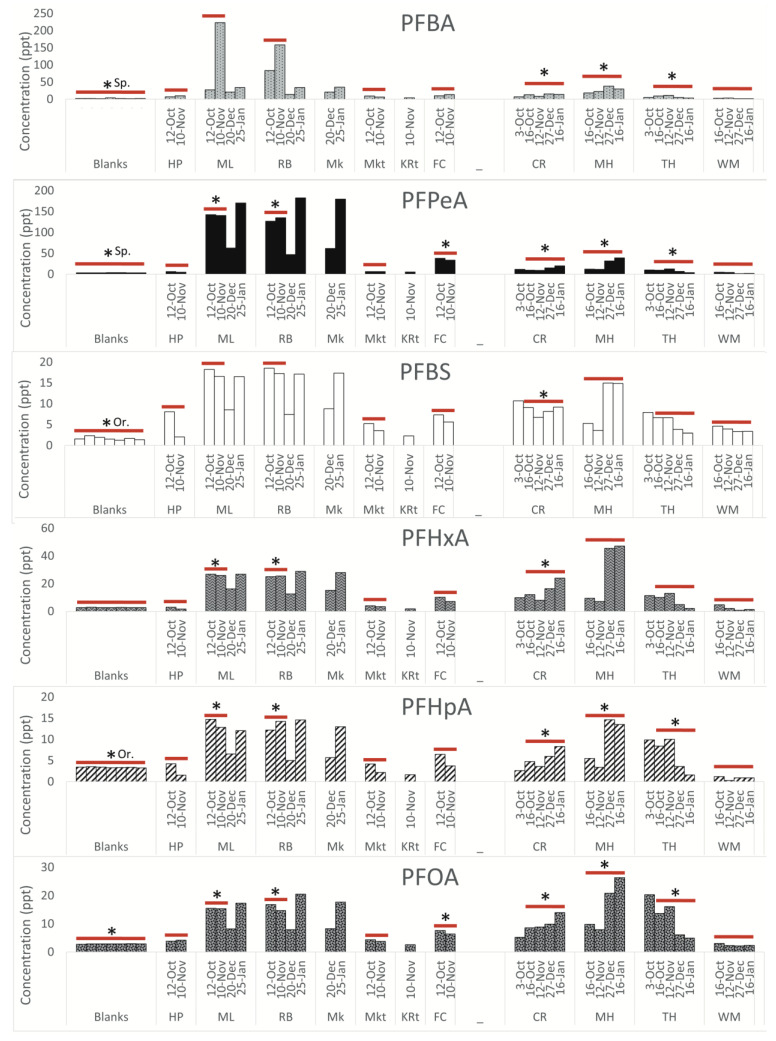
Legacy PFAS concentrations measured in each sample collected in Sampson County and Orange County. Procedural blanks are shown to validate that sample integrity was maintained. The LOQ for all compounds is 20 ppt. In Sampson County, fold change was calculated by averaging the concentration for samples collected on 12 October and 10 November and dividing by the average concentration of the reference site samples from 12 October and 10 November. In Orange County, fold change was averaged using samples collected on 16 October, 12 November, 27 December, and 16 January. Samples from these dates were selected for this calculation because we had samples from almost all of the sampling sites these dates. Samples included in these calculations are indicated by the red line. Statistical significance (*p*-value less than 0.05 when performing a *t*-test) is indicated by an asterisk. If the blanks were statistically significantly different than the reference site in only one county, the significant county abbreviation is included. The reference sites were HP in Sampson County and WM in Orange County. Dates are shown in day–month format. (Figure adopted from Walsh, 2020 [25]).

**Table 1 ijerph-20-06524-t001:** Upstream watershed area at each site.

County	Site	Upstream Watershed Area (Square Miles)	Latitude	Longitude
Orange	CR	2.58	35.980046	−79.065474
	MH	1.29	35.970772	−79.075416
	TH	0.43	35.959998	−79.09137
	WM	4.67	35.946031	−79.086835
Sampson	HP	10.91	35.013398	−78.448681
	ML	19.12	34.978726	−78.445647
	RB	19.93	34.964306	−78.444062
	MK	21.93	34.96129	−78.443881
	MKt	1.93	34.961405	−78.44232
	KRt	0.15	34.960034	−78.457296
	FC	128.5	34.927422	−78.465798

## Data Availability

Raw data were generated at the University of North Carolina, Gillings School of Global Public Health Mass Spectrometry Facility. Derived data supporting the findings of this study are available from the corresponding author, Courtney Woods, upon request.

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
