# Peer review of "Presence of Perfluoroalkyl Substances in Landfill Adjacent Surface Waters in North Carolina"

_ijerph, 2023, doi:10.3390/ijerph20156524_

Round 1
Reviewer 1 Report
The manuscript investigates the presence of PFAS in landfill adjacent water bodies. The technical English and formatting of the manuscript needs improvement, and more discussion needs to be added in the manuscript. Specific comments on the manuscript can be found in the attached annotated pdf.

Needs to be improved
Author Response
Point 1. Title needs correcting.
Response 1. Capitalization was corrected in the title and the spelling of “North Carolina” was corrected.
Point 2. These kind of emerging contaminants can also be remove via (bio)electrochemical technologies. Please explain that briefly in the introduction section. Refer to the following articles for further information.
Response 2. These technologies can be implemented but are not currently utilized by landfills. These technologies are utilized by some water treatment plants for public drinking water but the authors do not discuss water treatment plants and thought including information about these technologies could confuse readers about techniques utilized by landfills.
Point 3. Add latitude and longitude of sampling sites.
Response 3. Latitude and Longitude of sampling sites were included in Table 1
Point 4. Satellite images should be provided for figure 1.
Response 4. Satellite image was added to Figure 1 as Figure 1b.
Point 5. Explanation about duplicate or triplicate sampling analysis is needed
Response 5. The following explanation about repeated analysis (duplicates) on mass spectrometer for two samples was added to methods section: “Two duplicate samples were included to ensure consistency in the values obtained from the mass spectrometer.”
Point 6. Correct formatting on Figure 4 and 5.
Response 6. Formatting was corrected for caption of Figure 4 and 5.
Point 7. Expand short forms in table
Response 7. The authors felt that it was important for the table to be consistent with the plots and paper description and use abbreviations for the sampling sites.
Point 8. Conclusion is too long. Reduce to 150 words and only focus on the major findings
Response 8. Conclusion was shortened.
Reviewer 2 Report
Overall its a nice compilation of research done on the PFAS contamination into groundwater near two landfills i.e. SC and OC. Although manuscript is written well and substantiated with suitable researches, I have following recommendation in order to improve the manuscript and also to increase the readership.
1- The quality of figures is very low. I suggest author to provide high resolution pictures, and also to increase font size in the text of figures. It would be better if graph can be zoom in and color can be changed so that it can be visible/identical in black & white print.
2-If possible concise the material and method. Try to provide standard method followed, instead of explaining the washing/rising process repeatedly.
3- results should be Results (in section 3.)
4-How many samples were collected from each site (Number of replicates) is not mentioned.
5-Any statistical tool is not applied to justify the results and methods.
6-I suggest to change all graph and make them visible and apply standard deviation, wherever applicable.
English is fine, only a proof read is needed.
Author Response
Point 1. The quality of figures is very low. I suggest author to provide high resolution pictures, and also to increase font size in the text of figures. It would be better if graph can be zoom in and color can be changed so that it can be visible/identical in black & white print.
Response 1. Figures were remade to increase font size and improve readability in black and white. Figure 3 was divided into Figure 3a (Novel PFAS) and Figure 3b (Legacy PFAS) because when in black and white, the compounds were difficult to distinguish if all twelve PFAS species were included in one plot. Figures 4 and 5 were remade as well so that all colors and patterns on species remain consistent from Figure 3 and so that the legibility of the axes and labels are improved. Finally, resolution should be improved as well due to different uploading method.
Point 2. If possible concise the material and method. Try to provide standard method followed, instead of explaining the washing/rising process repeatedly.
Response 2. At the time this study was conducted, a standard method was still being developed for PFAS analyses. There is therefore not an adequate standard method that we could cite to explain our exact methods and to most accurately represent our methods we believe it important to include the steps taken for sample preparation for the mass spectrometer.
Point 3. results should be Results (in section 3.)
Response 3. Capitalization was changed.
Point 4. How many samples were collected from each site (Number of replicates) is not mentioned.
Response 4. Only one sample was collected from each site. A section explaining that two duplicates were included on the mass spectrometer to confirm consistency and reproducibility was added to the methods.
Point 5. Any statistical tool is not applied to justify the results and methods.
Response 5. As explained in the methods, t-tests were performed to determine if the fold change observed between sites was statistically significantly different. Plots were updated to show which sites had statistically significant differences for each compound. Language was also added to clarify that statistical significance was determined by a p-value of less than 0.05 from a t-test.
Point 6. I suggest to change all graph and make them visible and apply standard deviation, wherever applicable.
Response 6. While we do not have standard deviations available for the mass spectrometer data, we did remake the figures to improve legibility.
Reviewer 3 Report
Dear Authors,
The presented manuscript discusses the important problem of the presence of PFAS compounds, whose widespread practical use in everyday life is becoming a threat to the environment. In the manuscript, the authors confirm the penetration of PFAS compounds into the soil and water environment.
The design of the manuscript and the content presented is in line with the topic addressed. Its message and main idea is preserved. The following comments are indications suggesting a broader discussion in the fields:
1. In the introduction, I propose a more detailed discussion of the main types of PFAS and newly analyzed, pointing out the industries and products that we use most often in our daily lives and thus that can potentially have the greatest impact on our health. What proven concentrations and contact times have a real impact on the health and life of living organisms.
2. The Authors indicate that due to financial constraints there was no opportunity to analyze other parameters such as flow velocity, pH or specific conductance. I kindly ask to indicate, based on the Author's knowledge and review of the literature, how these parameters could have affected the interpretation of the presence of PFAS in the ecosystem. I have taken the liberty of posting two publications that may help answer the above questions :
a) Landfill leachate contributes per-/poly-fluoroalkyl substances (PFAS) and pharmaceuticals to municipal wastewater, Jason R. Masoner, Dana W. Kolpin, Isabelle M. Cozzarelli, Kelly L. Smalling, Stephanie C. Bolyard, Jennifer A. Field, Edward T. Furlong, James L. Gray, Duncan Lozinski, Debra Reinhart, Alix Rodowa and Paul M. Bradley
b) Occurrence and removal of poly/perfluoroalkyl substances (PFAS) in municipal and industrial wastewater treatment plants , Sibel Barisci; Rominder Suri
3. Authors research prompts to look for solutions to reduce PFAS in the legal and technological fields. Therefore I propose to indicate what treatment processes can be effective for PFAS removal from drinking water or wastewater.
Moreover, what ways of protecting the ground of landfills would be effective in reducing the transfer of PFAS into and out of the environment?
Suggestion:
a) PFAS National Environmental Management Plan Version 2.0 - January 2020 National Chemicals Working Group of the Heads of EPAs Australia and New Zealand
b) https://www.whitehouse.gov/wp-content/uploads/2023/03/OSTP-March-2023-PFAS-Report.pdf
4. Recently, there have been new guidelines and proposals for changes in the restriction of PFAS in drinking water. Kindly consider the latest suggestions, including concentration limits for selected types of PFAS. (https://www.epa.gov/sdwa/and-polyfluoroalkyl-substances-pfas).
Sincerely, Reviewer
Author Response
Point 1. In the introduction, I propose a more detailed discussion of the main types of PFAS and newly analyzed, pointing out the industries and products that we use most often in our daily lives and thus that can potentially have the greatest impact on our health. What proven concentrations and contact times have a real impact on the health and life of living organisms.
Response 1. Thank you for your comment. The main focus of this study was to show that waste containing PFAS that was disposed of in the landfill was traveling offsite into the environment. Follow up studies will investigate potential risk associated with exposure, however that is outside of the scope of this project.
Point 2. The Authors indicate that due to financial constraints there was no opportunity to analyze other parameters such as flow velocity, pH or specific conductance. I kindly ask to indicate, based on the Author's knowledge and review of the literature, how these parameters could have affected the interpretation of the presence of PFAS in the ecosystem. I have taken the liberty of posting two publications that may help answer the above questions :
- a) Landfill leachate contributes per-/poly-fluoroalkyl substances (PFAS) and pharmaceuticals to municipal wastewater, Jason R. Masoner, Dana W. Kolpin, Isabelle M. Cozzarelli, Kelly L. Smalling, Stephanie C. Bolyard, Jennifer A. Field, Edward T. Furlong, James L. Gray, Duncan Lozinski, Debra Reinhart, Alix Rodowa and Paul M. Bradley
- b) Occurrence and removal of poly/perfluoroalkyl substances (PFAS) in municipal and industrial wastewater treatment plants , Sibel Barisci; Rominder Suri
Response 2. This is a good point. For the purposes of this research, flow rate is the only really important parameter. Additional language about how flow rate can be used to estimate load and account for variations in hydrologic conditions has been added and language regarding pH and specific conductance was removed.
Point 3. Authors research prompts to look for solutions to reduce PFAS in the legal and technological fields. Therefore I propose to indicate what treatment processes can be effective for PFAS removal from drinking water or wastewater.
Moreover, what ways of protecting the ground of landfills would be effective in reducing the transfer of PFAS into and out of the environment?
Suggestion:
- a) PFAS National Environmental Management Plan Version 2.0 - January 2020 National Chemicals Working Group of the Heads of EPAs Australia and New Zealand
- b) https://www.whitehouse.gov/wp-content/uploads/2023/03/OSTP-March-2023-PFAS-Report.pdf
Response 3. Although there are current methodologies being developed to remove PFAS from waste water, this is outside the scope of this current study. The current goal of this paper would be to increase the testing performed for PFAS and to improve the way PFAS containing waste is managed by landfills.
Point 4. Recently, there have been new guidelines and proposals for changes in the restriction of PFAS in drinking water. Kindly consider the latest suggestions, including concentration limits for selected types of PFAS. (https://www.epa.gov/sdwa/and-polyfluoroalkyl-substances-pfas).
Response 4. These proposed standards have not yet been approved but would apply to drinking water, as opposed to environmental conditions. The purpose of this paper is to provide evidence that there is off-site migration of these contaminants and therefore there is a risk of these contaminants posing threat to nearby well water, however, we can not draw any conclusions about the potential concentrations in drinking water. Drawing these relationships between standards in drinking water and environmental concentrations would therefore be fairly arbitrary.
Round 2
Reviewer 1 Report
None
Good
Reviewer 2 Report
Authors have answered all the comments, and did changes wherever suggested.